# Avalanche Hazard Modelling within the Kráľova Hoľa Area in the Low Tatra Mountains in Slovakia

**Vladislava Košová** [1], **Mário Molokáč** [2,*] , **Vladimír Čech** [1] **and Miloš Jesenský** [3]

1   Department of Geography and Applied Geoinformatics, Faculty of Humanities and Natural Sciences, University of Prešov, Ulica 17. Novembra 1, 081 16 Prešov, Slovakia; vladislava.kosova@smail.unipo.sk (V.K.); vladimir.cech@unipo.sk (V.Č.)
2   Department of Geo and Mining Tourism, Faculty of Mining, Ecology, Process Control and Geotechnologies, Institute of Earth Resources, Technical University of Košice, Letná 9, 042 00 Košice, Slovakia
3   Belianum, Matej Bel University Press, Matej Bel University, Tajovského 51, 974 01 Banská Bystrica, Slovakia; milos.jesensky@umb.sk
*   Correspondence: mario.molokac@tuke.sk

**Abstract:** The aim of this work is a comprehensive assessment of the avalanche risk within the Kráľova hoľa area in the Low Tatra Mountains in Slovakia by the modeling of trigger areas, and the simulation of avalanche movements and their maximum impact using GIS and the RAMMS simulation model. Within the environment of geographic information systems, we created a layer of trigger areas using a digital elevation model and a vector layer of a land cover as input data. This layer was added together with the digital elevation model to the RAMMS simulation model, where cartographic outputs were created, focusing on snow cover height, avalanche flow speed, and pressure exerted by a falling avalanche. Based on these documents, we were able to develop an updated map of the avalanche cadastre of the examined area. In the given territory, we mapped a range of trigger areas within an area of 2.6 km$^2$ and the total range of avalanche run-outs within 14 interconnected areas. Of all the high mountains in Slovakia endangered by avalanches, this is the lowest range. The results are a suitable basis for the proper management and optimal use of the territory, which is part of Low Tatras National Park.

**Keywords:** avalanche risk; trigger area; RAMMS model; Kráľova hoľa; avalanche run-out

## 1. Introduction

Mountain area is generally a rare and fragile geosystem, which is frequently being attacked by humans, mostly through sports or other recreational activities. The ecological stability of such territories is often threatened by natural, anthropogenic, or natural-anthropogenic factors. One of these factors, which has a modeling impact as well, are avalanches. Snow avalanches represent one of the major natural dangers of mountain areas in the world, threatening settlements, infrastructure, natural resources, and last but not least, posing a great risk to human movement in alpine areas. Due to the high spatial variability and impermanent nature of the snow cover, predicting snow avalanches is extremely difficult, but very important to reducing their negative effects. The term snow avalanche refers to the sudden release of a snow mass on slopes due to the action of gravitation forces, which may also consist of the remains of loose rock blocks, soil, or vegetation [1]. For the purpose of distinguishing snow avalanches, a large number of classifications were created, with the De Quervain et al. [2] classification becoming accepted worldwide. The main classification factors of avalanches in this work are as follows: type of release, trajectory shape, and movement type.

The first scientific publications dealing with avalanches were published at the turn of the 20th century in the Alps, and then, half a century later, several American researchers began to address this topic as well. The work of the Swiss forester Coaza from the Swiss

Alps region is considered to be the first such publication, showing above all the signs of professional description [3]. Currently, the most comprehensive scientific work focusing on snow avalanches is "*The Avalanche Handbook*" [4], containing detailed information on the following subjects: the physical properties of snow; snow cover analysis; analysis of conditions; the causes of avalanches, their movement, and consequences; avalanche forecasting, prevention, and rescue; etc. Over the past decades, geographical information technologies have been used in the modelling, mapping, and visualization of avalanche terrain, which can clearly identify avalanche threats even within areas where there are no relevant records of avalanche events [5–8]. The first important step in the compiling of an avalanche cadastre is to identify potential trigger zones, what can be implemented by applying appropriate tools in the environment of geographic information systems [9–12]. Topographical relief parameters and, to a large extent, vegetation cover play a decisive role in the spatial expansion of trigger zones [13–16]). By the means of GIS tools, it is possible to determine the potential impact of avalanches and to prevent danger to human settlements and infrastructure. Statistical, deterministic, and conceptual models are now being used in practice to simulate the maximum impact of avalanches [13,17,18]. The first type of models are statistical (topographical) models, by which the maximum impact of avalanches is modeled, depending mostly on the topographical characteristics of the respective avalanche run-out [19–24]. The second type of models are deterministic (dynamic) models, predicting the speed, size, extent, and range of avalanches through numerical methods. When applying this model, it is necessary to obtain available input data, consisting of the height of the triggered snow mass and friction coefficients, which are difficult to determine without relevant avalanche records. It is the dynamic models transformed in the GIS environment that have been widely used by specialists in recent years [25–28]. The joint efforts of avalanche experts from around the world are directed at automating the process of mapping avalanche hazards. This can be achieved by analyzing images from the optical part of the high-resolution spectrum obtained by the remote sensing method [29–33].

The second largest mountain range in Europe is the Carpathian Mountains, which stretch over the territories of eight states. They reach their highest altitudes on the territory of the Slovak Republic (Gerlachovský štít 2654.4 m a.s.l.) in the geomorphological unit of the Tatras. However, the Low Tatra Mountains represent the largest Carpathian mountain range in Slovakia. In the Carpathian area, the first researcher to address the high mountain avalanche issue was Kňazovický [34–37], and later Lukniš [38], Midriak [39], and Milan [40–42]. The records of existing avalanche run-outs prepared by a pair of Slovak scientists, Kňazovický [36] and Milan [40], in a traditional form of field research became an empirical basis for the assessment of avalanche hazards of individual Western Carpathian mountains.

Considering more recent studies, the work of Hreško [43] presents one of the possible alternative methods of determining potential avalanche threats, with respect to previous empirical research and observations in the Tatra region. Other valuable contributions include works by: Hreško, Bugár [44]; Hreško, Bugár [45]; Rybár [46]; Barka [47]; Barka [48]; and Barka, Rybár [49]. Due to the higher degree of results relevance, M. Žiak [50] was able to develop the concept of the avalanche geographical information system, representing the updated digital version of the avalanche cadastre, by applying geo-information systems and remote sensing methods. Over the last decade, a large number of articles have been published which have made significant contributions to a better understanding of the processes associated with avalanche issues [51–57].

The aim of this work is a comprehensive assessment of the avalanche risk within the Kráľova hoľa area in the Low Tatra Mountains in Slovakia by the modeling of trigger areas, and the simulation of avalanche movements and their maximum impact using GIS and the RAMMS simulation model.

Only one avalanche accident was officially recorded in the studied area (1969). However, this does not mean that smaller avalanches do not occur here otherwise. The studied

area is remote from populated areas and relatively difficult to access. This leads to a lack of interest and to the absence of avalanche monitoring and recording in the studied territory. Rather, avalanche research is carried out in more accessible mountain areas of Slovakia such as High Tatras, Western Tatras, the western part of Low Tatras, as well as Great and Little Fatra. The studied territory also lacks the basic meteorological monitoring network which is present in other mountainous areas of Slovakia. For this reason, we decided to proceed with the assessment of the avalanche risk in the studied area in order to supplement the information on the overall avalanche risk of the whole of Slovakia. Similar research has not yet been carried out in the studied territory.

Before initiation of the research, four research questions (hypotheses) were identified:

1. Due to the nature of the territory of the Kráľova hoľa area (relief conditions associated with the extent of the quaternary glaciation, land cover, morphometric conditions, etc.), we assumed there would be a greater occurrence of trigger areas and areas with a higher degree of avalanche threat on the northern slopes of the studied area.
2. Compared to the various other studies carried out in Slovakia on a similar topic, we assumed there would be a lower area of trigger zones within the Kráľova hoľa area than in other Slovak mountains threatened by potential avalanche activity.
3. We assumed there would be a combination of several factors entering into the assessment of avalanche endangerment—not just one dominant factor.
4. We assumed that the design process of the avalanche cadastre based on results from the RAMMS model was an appropriate tool for optimizing the potential use of the territory by humans and preventing possible risks arising from avalanche endangerment.

## 2. Materials and Methods

### 2.1. Study Area

#### 2.1.1. Geological and Geomorphological Conditions

The study area of Kráľova hoľa covers an area of almost 35 km$^2$. It is situated in the easternmost part of the geomorphological sub-region of the Kráľova hoľa area of Low Tatras and the geomorphological unit of Low Tatras in the middle Slovakia (Figure 1). The stated area occupies the peak areas and adjacent slopes of the main ridge of the Low Tatra mountains (Figure 2), stretching from Ždiarske sedlo (1473 m a.s.l.) through Bartková (1790 m a.s.l.), Orlová (1840 m a.s.l.), Stredná hoľa (1876 m a.s.l.), and Kráľova hoľa (1946 m a.s.l.) to Kráľova skala (1690 m a.s.l.). The maximum length of the territory in the direction from west to east is 10 km, and the maximum width in the direction from north to south is 7 km. The highest point of the examined area is Kráľova hoľa with an altitude of 1946 m a.s.l., and the lowest point of the territory (1007 m a.s.l.) is located in the area of the Šumiacka dolina valley, where the watercourse Šumiacky potok exits the studied area. The studied area belongs to the cadastral territory of four municipalities—Liptovská Teplička, Pohorelá, Šumiac, and Telgárt.

The geological structure of the main ridge of the eastern part of the Kráľova hoľa area of Low Tatras is mainly composed of igneous and metamorphic rocks, represented by boulders, phyllites, gneisses, and para-gneisses of the Vepor Belt of the Inner Western Carpathians [58]. Spatially, the second most widespread lithological units, extending on the northeastern and eastern slopes of Kráľova hoľa (1946 m a.s.l.) and on the edges of glacial cirques, are deluvial-proluvial sediments and slope clays as well as breccia from the Quaternary period. The aforementioned geological period also includes fluvial alluvial clays, which are located in the southeastern part of the studied area, along the Šumiacky potok watercourse. The remnants of the Pleistocene glaciation of the Veľký Brunov, Holičná, and Bezmenná valleys, located northeast of the Bartková (1790 m a.s.l.), are fluvial and glaciofluvial sandy gravels and glacigenic moraine sediments deposited in the bottom parts of the aforementioned glacial cirques. In addition to the above lithological units, we observed the presence of metamorphic conglomerates, sandstones, slates and arkoses from the Permian in the northeastern part of the studied area, at the northeastern foot of the Tri kopce (1507 m a.s.l.) [58].

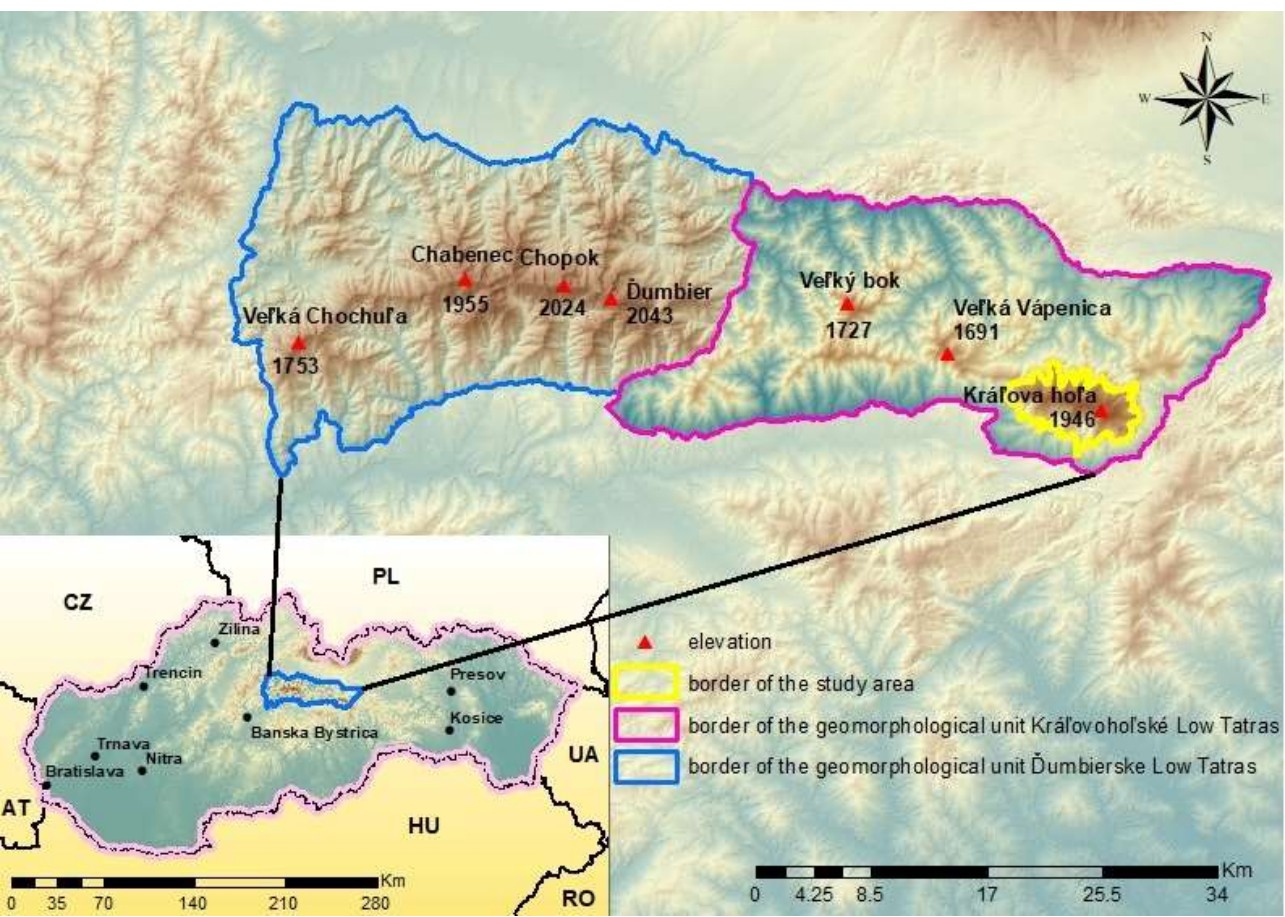

**Figure 1.** The location of the studied area. Source DEM: The Geodesy, Cartography and Cadastre Authority of the Slovak Republic.

Several authors [59–70] have already carried out a comprehensive evaluation of individual aspects of the Pleistocene glaciation in the region of the Kráľova hoľa area of Low Tatras. The existence of glacial cirques at the ends of selected valleys and frontal moraines occurring at different altitudes confirms the Pleistocene glaciation of that area. Würm glaciation can only be observed in the valleys of Veľký Brunov, Holičná, and a nameless valley located in a northeasterly direction from Bartková (1790 m a.s.l.), all of which are located on the north side of the ridge of the studied area. An average altitude of the bottom part of these glacial cirques is approximately 1580 m a.s.l. The slight asymmetry in the transverse profiles of the glacial cirques reflected by steeper left slopes with eastern and southeastern orientations is caused by climatic factors. The absence of typical rock steps separating the cauldron parts of the valley with the lower, U-shaped parts is a result of the relatively short duration of the Pleistocene glaciation. The longest glacier within the studied area, with a length of 2.9 km, was located in the valley complex Veľký Brunov, representing the most developed glacial-type valley in the region of the Kráľova hoľa area of Low Tatras. The steep rock walls surrounding the glacial cirques show signs of local gravitational deformation, which can be observed especially in the final sections of the nameless valley lying between Orlová (1840 m a.s.l.) and Bartková (1790 m a.s.l.). The frontal moraines of maximum glaciation (W2) were formed in all of the mentioned valleys (in the Valley of Veľký Brunov at an altitude of 1180–1250 m a.s.l. and in the two other valleys at an altitude of 1310–1340 m a.s.l.), and when comparing the altitudes of glacial cirques and frontal moraines of maximum glaciation (W2), the following rule applies: the lower the bottoms of the glacial cirques lie, the higher the frontal moraines are to be found. The final sections of the valleys, lying above the level of the snow line during the last

glacial period, were generally remodeled by nivation processes to the so-called nivation cirques (caroids), defined as shallow depressions with flat bottom parts, which narrow down in the direction of the slope wedge-wise and pass through steeper sections into straight, closed areas and at the beginning of trough valleys created by linearly acting erosion. In the valleys remodeled by nivation, there are rock glacier accumulations in tongue form dating from the peak Würm period, whose steep frontal parts function as transverse barriers are thus cause the valley parts to expand. Accumulations resembling rock glaciers, containing a large number of fine-grained, predominantly alumina, sandy sediments, represent accumulations of mud flows that often penetrate into lower parts of valleys than moraines or rock glaciers from the last glacial period do.

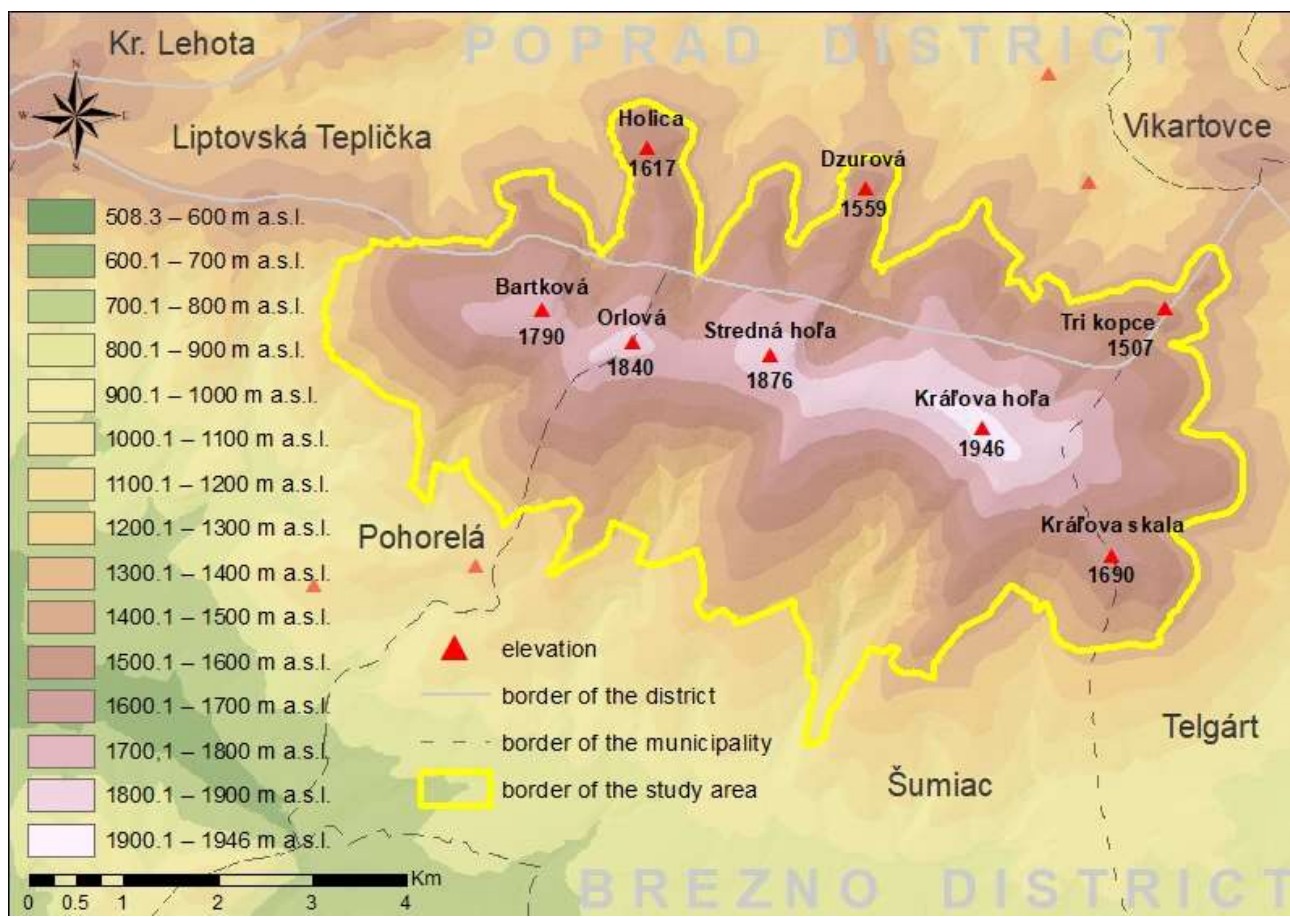

**Figure 2.** Hypsometric conditions of the studied area within the administrative boundaries. Source: The Geodesy, Cartography and Cadastre Authority of the Slovak Republic.

### 2.1.2. Climatic and Vegetation Conditions

Climatic conditions within the studied area depend mainly on altitude, slope exposure, terrain configuration, season, and atmospheric circulation. We obtained relevant information about the snow conditions of the studied area by analyzing data from the nearest meteorological station, located approximately 40 km west, below the Chopok peak (2024 m a.s.l.). Based on the climatic classification of the territory of the Slovak Republic [71], the studied area falls into a cold climatic zone, within which we recognize three basic regions with high humidity: cold mountain region, relatively cold mountain region, and moderately cold region. The coldest region is characterized by average July temperatures not exceeding 10 °C, annual precipitation ranging from 1200 to 2130 mm, the number of days with snow cover ranging from 180 to 200 days, and with the average seasonal maximum snowfall at a value of 140 to 160 m. This type of region occupies the

top parts of the Low Tatras ridge, stretching from Ždiarské sedlo (1473 m a.s.l.) through Bartková (1790 m a.s.l.), Orlová (1840 m a.s.l.), Stredná hoľa (1876 m a.s.l.), and Kráľova hoľa (1946 m a.s.l.) to Kráľova skala (1690 m a.s.l.). The relatively cold mountain region is defined by average July temperatures ranging from 10 °C to 12 °C, total precipitation ranging from 1000 to 1400 mm, the number of days with snow cover from 160 to 180, and a maximum snowfall level of 120 to 140 m, while the region includes areas with an altitude of at least 1300 m a.s.l. The third, moderately cold region, with an average July temperature of 12 °C to 16 °C, annual precipitation of 800–1100 mm, a maximum number of days with snow cover set at 160, and a maximum snowfall level ranging from 100 to 120 m, is located in the lowest parts of the study area [71]. In the wider vicinity of the area of interest, the air circulation in the direction of the meridian prevails, while the ratio of circulations from the north and south is more or less balanced. When flowing in a parallel direction, the circulation from the west significantly prevails. The analysis of the average annual wind speed shows that the windiest areas of the studied territory are the top parts of the Low Tatras ridge from about 1500 m a.s.l., where the annual wind speed ranges from 6 m/s to 10 m/s. The windiest place in Slovakia is the peak of Chopok, where the average annual wind speed reaches 10 m/s. In lower, more forested areas, the average annual wind speed decreases between 5 and 6 m/s [72].

When characterizing the vegetation cover, it is necessary to account for the significant vertical fragmentation of the studied area, and therefore it was necessary to divide the area into three vertical vegetation zones. The first of these is a vertical zone of spruce forest communities, extending from altitudes of 1002 m a.s.l. up to the upper border of the forest, lying at an altitude of approximately 1500 m a.s.l., populated mainly by Norway spruce, mountain maple, and European larch. These forest communities are currently in poor condition due to wind and bark beetle calamities and excessive logging. The second zone, reaching an altitude of approximately 1700 m a.s.l., is the vertical zone of scrub, the main constituents of which are Swiss stone pine, mountain cranberry, European blueberry, and Siberian cypress. All localities situated above the border of 1700 m a.s.l. belong to the vertical zone of alpine meadows and are covered with swamp grasses. The level of the upper forest boundary as well as the scrub forest zone were significantly affected by grubbing-up and burning by shepherds during the Wallachian colonization.

### 2.2. Data Sources and Pre-Processing

The overall accuracy of the final results regarding the avalanche risk to the examined area depends primarily on the detail of the input data. In the basic input data we included a digital elevation model, which is required in the analysis of topographical factors which enter the process of trigger zone formation and then subsequent simulations of snow avalanche movements. For the purposes of this paper, we used the digital elevation model 3.5 in raster form in TIFF format in the S-JTSK coordinate system at a resolution of 10 m/pixel, published on the official website of The Geodesy, Cartography and Cadastre Authority of the Slovak Republic. Another important form of input data was the vector layer of land cover which had to be independently developed due to the absence of relevant data, using up-to-date aerial images with a resolution of 1 m/pixel, obtained from The Geodesy, Cartography and Cadastre Authority of the Slovak Republic and the National Forestry Centre, and subsequently verified by field research.

### 2.3. Trigger Zones Modeling

The modelling of potential trigger zones is based on the need to identify specific locations where the risk of triggering the released snow mass is greatest. Several authors have developed statistical models determining these zones in the conditions of the Western Carpathians. J. Hreško [43] is considered to be a pioneer in this area for developing the

equation calculating breakaway zones based on statistical analysis of the topographical properties of registered avalanches as follows:

$$Av = (Al + Ex + Fx + S) \times Rg$$

Av (avalanche hazard), Al (altitude factor), Ex (exposure factor), Fx (plan curvature factor), S (slope factor), Rg (surface roughness factor).

Comparing this equation with the avalanche cadastre by two authors, Bárka and Rybár [49], results in its modification into the following form:

$$Av = (Al + Ex + Fx) \times S \times Rg$$

Av (avalanche hazard), Al (altitude factor), Ex (exposure factor), Fx (plan curvature factor), S (slope factor), Rg (surface roughness factor).

Aiming at more relevant and accurate calculation results of potential trigger zones, Biskupič [73] added another variable in the form of a profile curvature factor to the equation:

$$Av = (Al + Ex + Fx + Fy) \times S \times Rg$$

Av (avalanche hazard), Al (altitude factor), Ex (exposure factor), Fx (plan curvature factor), Fy (profile curvature factor), S (slope factor), Rg (surface roughness factor).

Altitude has a fundamental influence on the formation and character of avalanche processes. Altitude, together with the direction of mountains, plays an important role in climatic and vegetation zoning. As the altitude increases, the average temperatures decrease proportionately, and precipitation increases. As a result, the thickness of the snow cover increases at higher altitudes.

Conversely, due to higher temperatures, the snow cover is thinner and much more sensitive to temperature fluctuations at lower altitudes. Changes in the metamorphic processes of the snow cover and its physical and mechanical properties are the result of an increase in altitude.

We created a raster layer of the altitude factor by reclassifying the digital relief model into three intervals according to Table 1.

The effect of slope exposure on avalanche processes is not clear. The effect of exposure on avalanches is influenced to a large extent by whether the slope is facing towards the direction of the prevailing winds or towards the direction of the sun's rays. The orientation of the slope towards the direction of the prevailing winds is reflected in the varying placement of snow cover. Strong winds during intense and long-lasting snowfall are often the cause of avalanches on leeward slopes. The exposure has a greater effect mainly due to the unequal effects of sunlight on avalanche processes. The different effects of solar radiation affect the temperature regime of snow layers and their metamorphic processes.

The varying stratification of the snow cover, and the varying formation of its mechanical and physical properties, are caused by the lasting effect of different doses of sunlight. The mechanical properties of snow are impaired by the overheating of snow layers on the southern and southwestern slopes, which is a very common phenomenon.

This is how the trigger zone and the subsequent avalanche is formed. The exposure factor takes into account the direction of the slope. The influence of solar radiation and the influence of wind on the deposition of snow cover were taken into account by including this factor in the model. In general, this is most visible on the southern and eastern slopes.

Slopes with a southeasterly (135°) to westerly (270°) direction were classified as risky slopes according to the percentage. The remaining exposures enter the model with lower values. Raster directions correspond to points of the compass, calculated from a digital relief model by the Aspect module. The new layer was then reclassified according to Table 1.

**Table 1.** Values of variables entering the calculation of trigger zones [51].

| Altitude (m a.s.l.) | Altitude Factor (Al) | Profile Curvature (in m$^{-1}$) | Profile Curvature Factor (Fy) | Plan Curvature (in m$^{-1}$) | Plan Curvature Factor (Fx) |
|---|---|---|---|---|---|
| below 1300 | 0 | (−4)–(−0.2) | 1 | 4–0.2 | 1 |
| 1300–1500 | 1 | (−0.2)–0.2 | 1 | 0.2–(−0.2) | 1 |
| over 1500 | 2 | 0.2–0.5 | 1 | (−0.2)–(−0.5) | 1 |
| | | 0.5–4 | 0.5 | (−0.5)–(−4) | 0.5 |
| **Exposure** | **Exposure factor (Ex)** | **Slope (in degrees)** | **Slope factor (S)** | **Type of land cover** | **Roughness factor (Rg)** |
| NW | 0.4 | (0–10)°, (70–90)° | 0 | forest (coniferous, broadleaf, mixed) | 0.5 |
| NE | 0.5 | (10–19)°, (60–70)° | 0.4 | incoherent forest with mountain pine, thick boulder debris and slopes with smaller blocks | 1.2 |
| E | 0.7 | (19–25)°, (55–60)° | 0.8 | shrub cover | 1.4 |
| N | 0.8 | (25–30)°, (50–55)° | 1.2 | sparse forest (grass and trees) | 1.5 |
| SW | 1 | (30–35)°, (45–50)° | 1.6 | continuous mountain pine, slopes with bedrock protrusions up to 50 cm | 2.5 |
| SE | 1.5 | (35–45)° | 2 | grass with incoherent mountain pine, small fragmentary debris slopes | 2.8 |
| W | 1.7 | | | grassland, rock seas, rock plates | 3 |
| S | 2 | | | | |

Annotation: The higher the value of the factor, the higher the risk of trigger zone formation, except for the values of plan and profile curvature.

The curvature of the relief plays an important role in the formation of avalanches. Horizontal (or planar) curvature is actually the curvature of the contour line. Gulliers and so-called corners display concave shapes, and terrain ridges display convex shapes. These are examples of horizontal curvature. Inflation accumulates snow masses in concave shapes which results in an increased probability of avalanches. The force of the wind acts on convex terrain shapes and snow is carried from them. The height of the snow cover is smaller compared to concave shapes and this reduces the probability of avalanches. Concave shapes have an accelerating effect on avalanches and convex shapes tend to slow down avalanches.

Profile curvature is curvature in the direction of the relief slope. We distinguish between convex and concave shapes, both in horizontal curvature and in profile. There is tension in the snow cover due to the convex shape of the relief which results in an increased probability of an avalanche. The convex form constantly increases the slope which also increases the tension in the snow. There is a reduction in the slope in the concave shape of the terrain which causes a reduction in tension in the snow layers or the slowing down of an already formed avalanche. Concave surface shapes present a risk of avalanches due to their ability to accumulate wind-borne snow. We can establish whether a slope is concave or convex by DMR analysis using the Curvature tool. We reclassified the calculated layers of planar and profile curvature according to Table 1.

Slope steepness is one of the key factors in determining avalanche danger. It is prevailing opinion among the lay public that avalanches are formed only on steep slopes. However, the truth is that most avalanches occur on slopes that have an incline of 35° to 45°. This range represents the zone where the balance between the cohesion of the snow layers and the effect of gravitational force is disturbed. There is not enough accumulation of snow masses on slopes with an incline greater than 60°, because the snow is not able to stay there. Conversely, if the angle is less than 25°, the slope is not steep enough for the avalanche to be released. The data on critical slope angle vary considerably and have more or less local validity. It is not expected in the future that there could be a global uniform value of the critical slope angle for the formation of an avalanche.

The whole process of avalanche formation depends on local terrain and climatic conditions and on the height and character of the snow cover.

The slope conditions in the trigger zone are decisive for the formation of an avalanche. These ratios determine whether the snow layers are torn and what speed the subsequently torn-off snow masses reach.

The dynamic effect in the collecting (accumulation) area or on the avalanche path is also affected by this. Creeping, rolling, and slipping predominate on small slopes. Bouncing or air movement is achieved at large slopes. We derived the slope layer by degrees using the Slope Module from a digital relief model which represents the slope conditions in the area of interest. We then reclassified this layer to create the resulting slope factor layer.

Landscape structure located directly under the snow is another factor that significantly influences the risk of an avalanche.

Vegetation plays an important role in the consolidation and stabilization of snow. The subsoil under the snow layer, such as grassland or stone rubble, allows the formation of full-depth slab avalanches. The formation of avalanches is very limited in dense vegetation and the forest can even capture an already released avalanche. Natural obstacles, such as large boulders, scrubs, or the remains of dead trees are avalanche inhibitors. If the snow cover is so deep that it overcomes these obstacles, then their inhibitory effect is lost. The surface roughness factor is derived from the land cover layer. This layer in the vector representation contains polygons of individual types of land cover. We converted this layer to a raster representation with the same spatial reference frame as the DMR using the Feature to Raster tool. The value of the surface roughness factor was assigned to each type of land cover by reclassification. Grasslands have the highest value and forest is the lowest.

The most important input data for the modeling of potential trigger zones are the digital elevation model (DEM), on which basis we are able to derive the required parameters

(altitude, orientation, slope, profile, and plan curvature) of the relief and the layer of land cover, providing important knowledge regarding the surface type. A necessary step before the actual implementation of the trigger zone calculation is a reclassification of all factors that are defined within the above equation. Sites with a higher probability of avalanche trigger in terms of altitude, exposure, slope, and surface were associated with higher factor values. The presented values of individual factors were defined in [51] according to long-term observations, statistical analysis of recorded avalanche situations, and practice (Table 1).

The resulting values of the trigger zones calculation should be reclassified according to the degree of danger of snow mass release up to the four basic classes (Table 2).

**Table 2.** Final reclassification.

| The Result of Equation Av | Danger of an Avalanche Trigger |
|:---:|:---:|
| 0–15 | small |
| 15–22.5 | medium |
| 22.5–30 | high |
| 30–36 | very high |

*2.4. Simulation of Avalanche Run-Outs and Avalanche Range*

2.4.1. Basic Characteristics of the RAMMS Model

RAMMS is highly developed two-dimensional numerical simulation model, used to calculate geophysical mass movements from initiation to its maximum reach in three-dimensional space [74]. The authorship of the model known as Avalanche, Debris Flow and Rockfall belongs to a development team of avalanche specialists associated in the WSL Institute for Snow and Avalanche Research, the SLF based in Davos, Switzerland, and the WSL Swiss Federal Institute for Forest, Snow and Landscape Research, based in Zurich, Switzerland. By means of this model, it is possible to indicate the movements of snow and stone avalanches, shallow landslides, as well as debris flows in terms of quantity, speed, and pressure of the mass. The interpretation of the results helps to design preventive, technical, and biological protection measures, including the installation of winter bar markings, warnings, braking barriers, as well as forest planting. After careful calibration and verification of the model associated with the availability of relevant input data, this model can also be applied in the conditions of the Western Carpathians.

In order to simulate the snow mass movement on the slope, the physical model of RAMMS::AVALANCHE uses Voellmy's law of friction. The model defines two basic types of terrain frictional resistance—dry (Coulomb) friction type (coefficient μ), in the RAMMS model referred to as Mu, and the coefficient of resistance depending on the quadrant of speed (coefficient $\xi$), measured in units (m/s$^2$), called Xi. The resulting friction resistance ($S$), indicated in units of pressure (Pascal), is calculated based on of the following relationship [74]:

$$S = \rho h g \cos(\phi) + \frac{\rho g u^2}{\xi}$$

$\rho$ (flow density), $h$ (flow height), $g$ (gravitational acceleration), $\phi$ (slope angle), $u$ (flow speed).

This formula has been used in many Swiss scientific studies dealing primarily with simulations of snow avalanche movements, but after a consistent calibration of frictional and global parameters, its application is also possible in the Western Carpathians. The main disadvantage of the Voellmy's model is a certain inaccuracy in the simulation of smaller avalanches, in which run-outs are too short for the accumulated snow mass not to stop directly at its trigger point. Due to the occurrence of smaller avalanches within the studied area it was necessary to increase the values of friction parameters (Mu and Xi) for modelling purposes.

2.4.2. Input Data for the RAMMS Model

The mandatory input data for the launch of RAMMS software are essentially two basic components—a digital elevation model and a polygon layer of trigger zones. The first of these data provides us with basic information regarding the terrain characteristics of the studied area, whereas the accuracy of the final simulation depends precisely on the resolution of this raster layer. In most cases, we used a digital relief model with a resolution of 5 m, however, in a thorough simulation, a 2 m raster can also be used. Nevertheless, using this spatial resolution significantly increases the time requirements for modelling. The vector layer of trigger zones also enters the process of creating the resulting model. The layer is defined by polygon formations of irregular shape showing the size and location of the initial snow mass which is descending towards the foot of the slope, while the volume of the snow mass increases. Another important value determining the accuracy of the simulation of avalanche movements is the height of the avalanche trigger point, which was set at 0.5 m following statistical analyses of avalanche events.

The RAMMS model has the option of an automatic calculation of the total surface friction through the input data (altitude, slope of relief, curvature of relief, afforestation of the territory). In addition to the aforementioned method of calculating the surface friction parameters, there is the option of adding an additional vector layer, containing more detailed information regarding the parameters of the surface friction. Before modeling itself, it is possible to modify these parameters by transforming the values of the Mu and Xi coefficients.

## 3. Results

### 3.1. Results of Triggers Zones Modelling

The first important step in the process of creating trigger zones in ArcGIS 10.2 was to obtain a digital relief model. Based on the model, using the Aspect, Curvature and Slope tools, we developed raster format layers, the values of which needed to be reclassified using the Reclassify tool according to Table 1. The Raster Calculator tool entered the values into the required formula for the calculation of potential trigger zones according to [73]. Then we were able to create a raster format layer, showing the locations of potential risk of triggering snow avalanches (Figure 3). The resulting equation values were reclassified according to the degree of avalanche hazard into four basic intervals (Table 3). Based on the calculations we were able to conclude that 92.44% of the total area of the territory is classified in the category with a small risk of triggering a snow avalanche, 5.87% with a medium risk, 1.53% with a high risk, and 0.16% with a very high risk. The complete results, together with a conversion to area units ($km^2$), are summarized in Table 3. Based on the results, it can be observed that the ridge parts of the territory show a low potential for snow mass triggering, whereas steeper slopes primarily surrounding glacial cirques and covered with grassland were correctly classified as areas with a high or very high risk of triggering snow avalanches. These are specifically areas of glacial cirques at the end of the valleys of Veľký Brunov, Holičná, and in the nameless valley northeast of the Bartková (1790 m a.s.l.) on the north side of the ridge. On the southern slopes of the examined area, there are areas with a high and very high potential for snow avalanches, located at the ends of the Lukačíková, Ždiarna, and Šumiacka valleys. Perpendicular rocky sections within the individual glacial circuits with a present spontaneous fall of rock fragments show an extremely high potential for avalanche triggering.

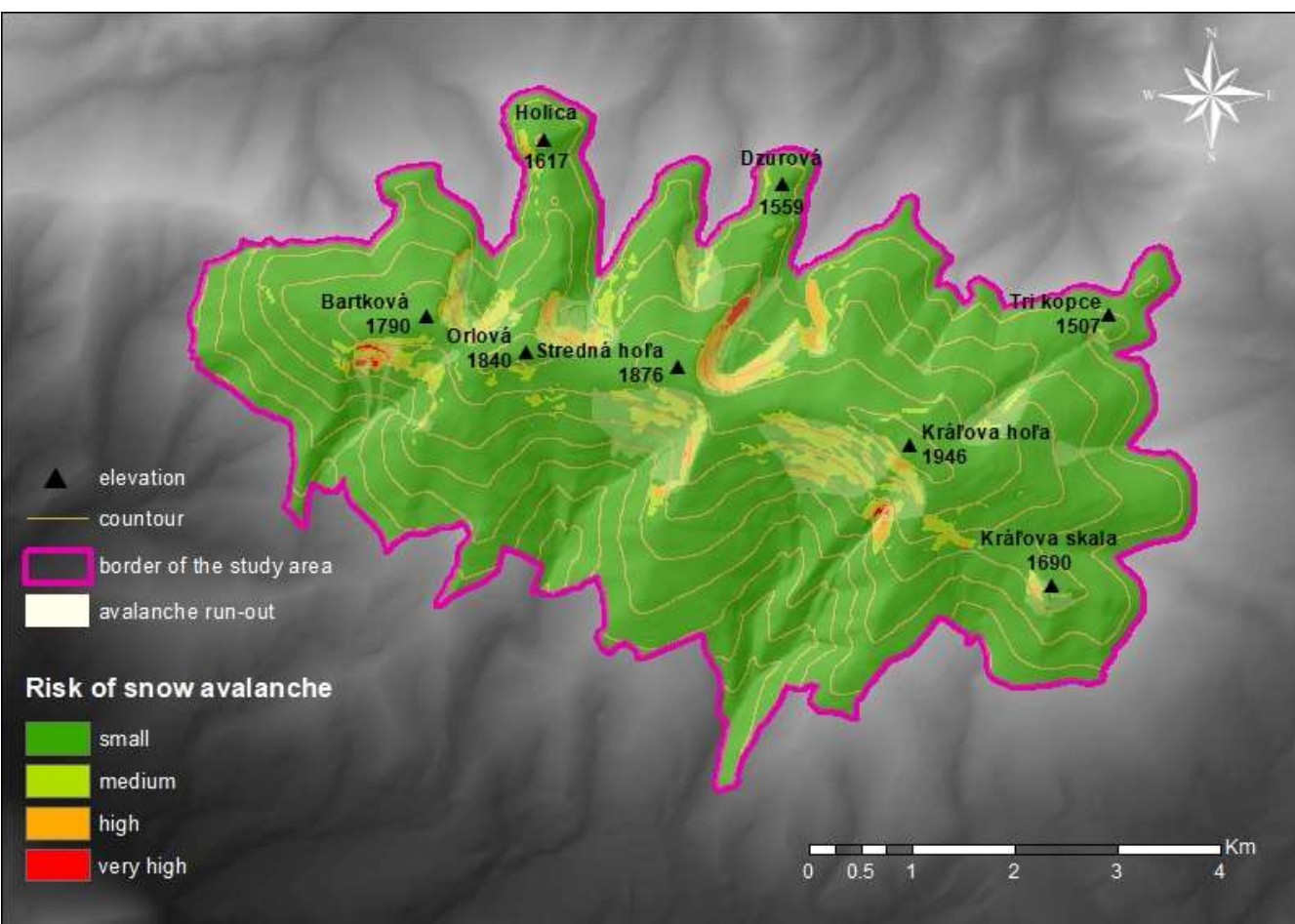

**Figure 3.** The avalanche release potential within the studied area compared with records in the existing avalanche cadastre.

**Table 3.** Proportional outcome of territories endangered with avalanche threat.

| Risk of Snow Avalanche | Percentage of the Studied Area * | Expanse of Territory |
|:---:|:---:|:---:|
| small | 92.44% | 32 km$^2$ |
| medium | 5.87% | 2 km$^2$ |
| high | 1.53% | 0.5 km$^2$ |
| very high | 0.16% | 0.1 km$^2$ |

* This is the percentage of the area with a given degree of risk out of the total studied area. (Total studied area has 34.6 km$^2$).

### 3.2. Results of Avalanche Run-outs and Range Simulations Using the RAMMS Model

By analyzing the input data, the RAMMS model is able to provide us with comprehensive information regarding the snow flow height, the speed of avalanche flow, and the pressure exerted during the avalanche fall.

Considering the first factor, namely the maximum snow flow height (Figure 4), it can be concluded that the maximum value of this factor is 16.91 m. This value was found in the Šumiacka Valley at an altitude of approximately 1300 m a.s.l. in the accumulation part of the valley, where several side gutter casts emerge. The highest values of snow flow height (8.5–16.91 m) were measured in the accumulation parts of the Lukačíková, Ždiarna, Šumiacka, and Havraník Valleys at altitudes from 1100 to 1400 m a.s.l. The lowest values of the snow flow height were located on steep grassy slopes just below the ridge part at altitudes from 1450 m a.s.l. to 1930 m a.s.l.

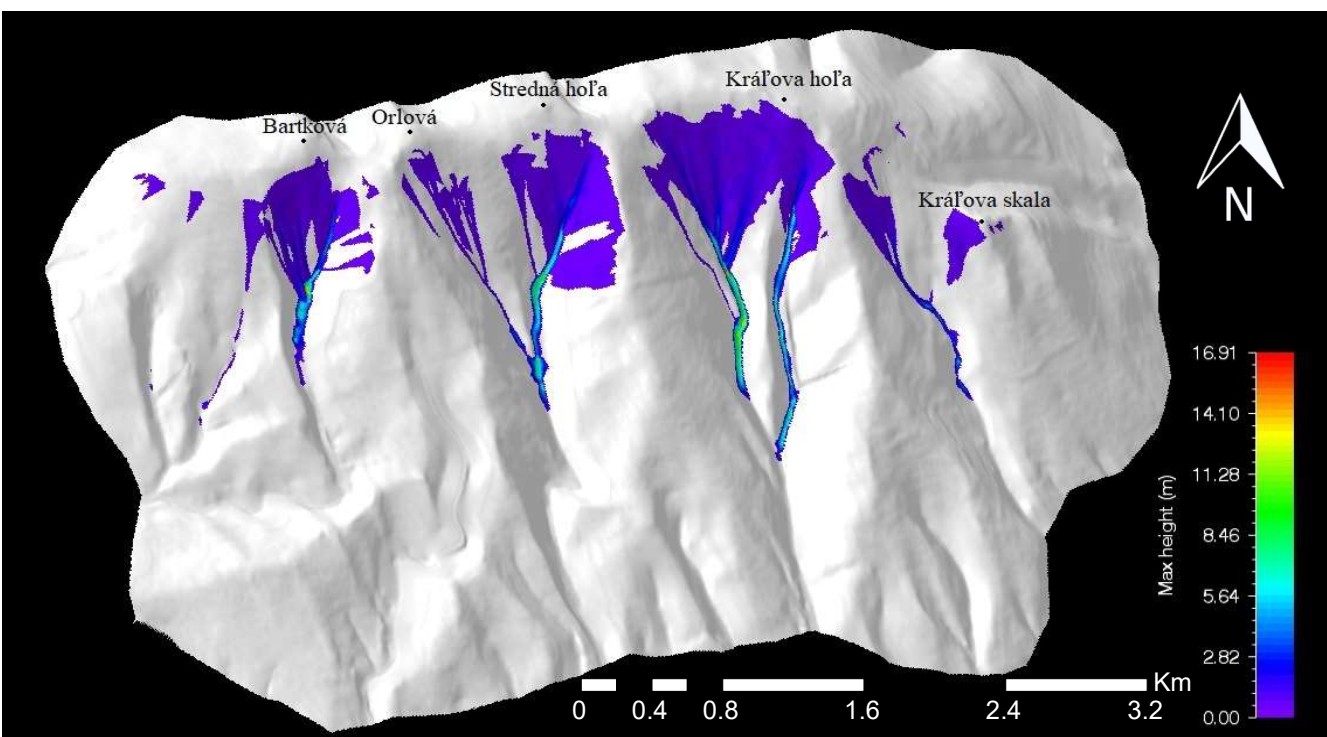

**Figure 4.** Maximum snow flow height (southern slope).

The second factor processeds within the RAMMS model is the maximum speed of avalanche flow (Figure 5). High values in the range from 28 to 34 m/s were measured in the area of narrow steep gutter casts within the valley complexes of Lukačíková, Ždiarna and Šumiacka. The altitude of these sections ranges from 1380 m a.s.l. to 1580 m a.s.l. The vast majority of avalanche speeds in the examined area lies in the range of 15 to 28 m/s.

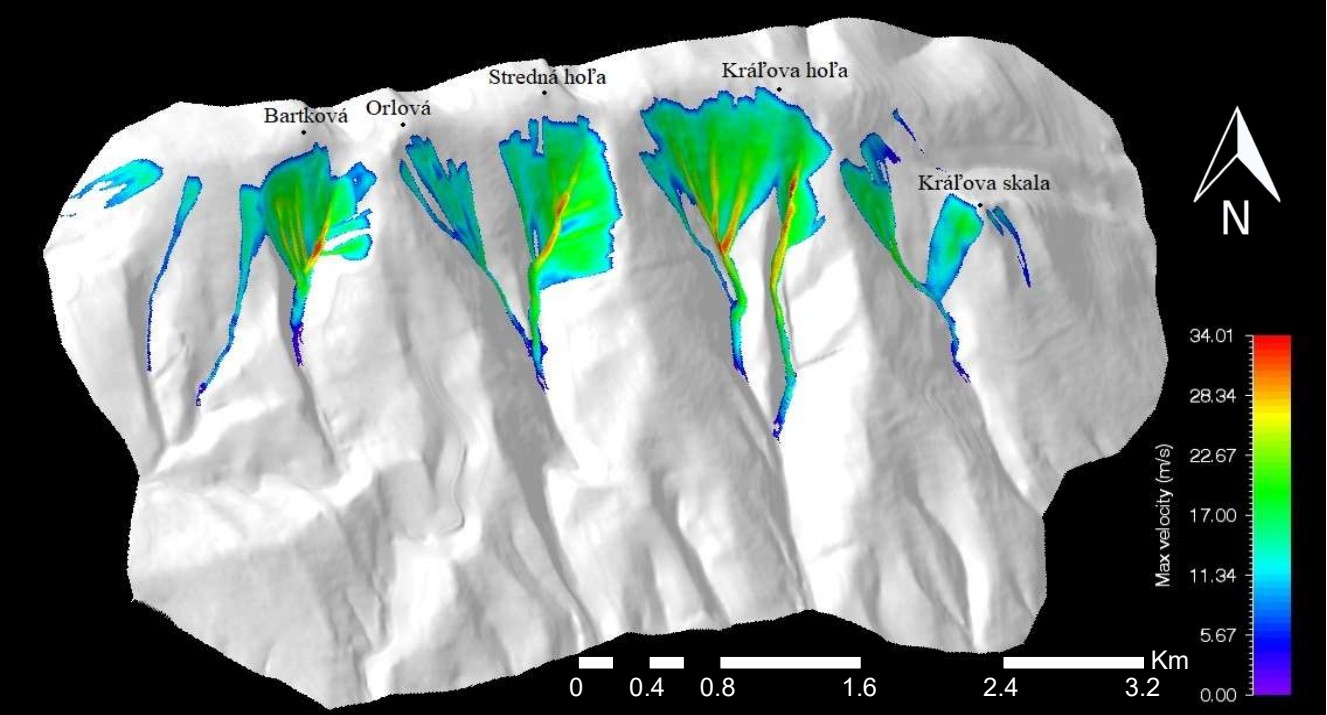

**Figure 5.** Maximum speed of avalanche flow (southern slope).

The average pressure exerted in an avalanche fall (Figure 6) in the examined area shows values of between 25 and 130 kPa. There is a significant increase in the pressure observed in the area of narrow steep gutter casts facing the valleys Lukačíková, Ždiarna, and Šumiacka, which culminates in the accumulation area where several side gutter casts emerge. The maximum pressure value obtained in the examined area is 347 kPa.

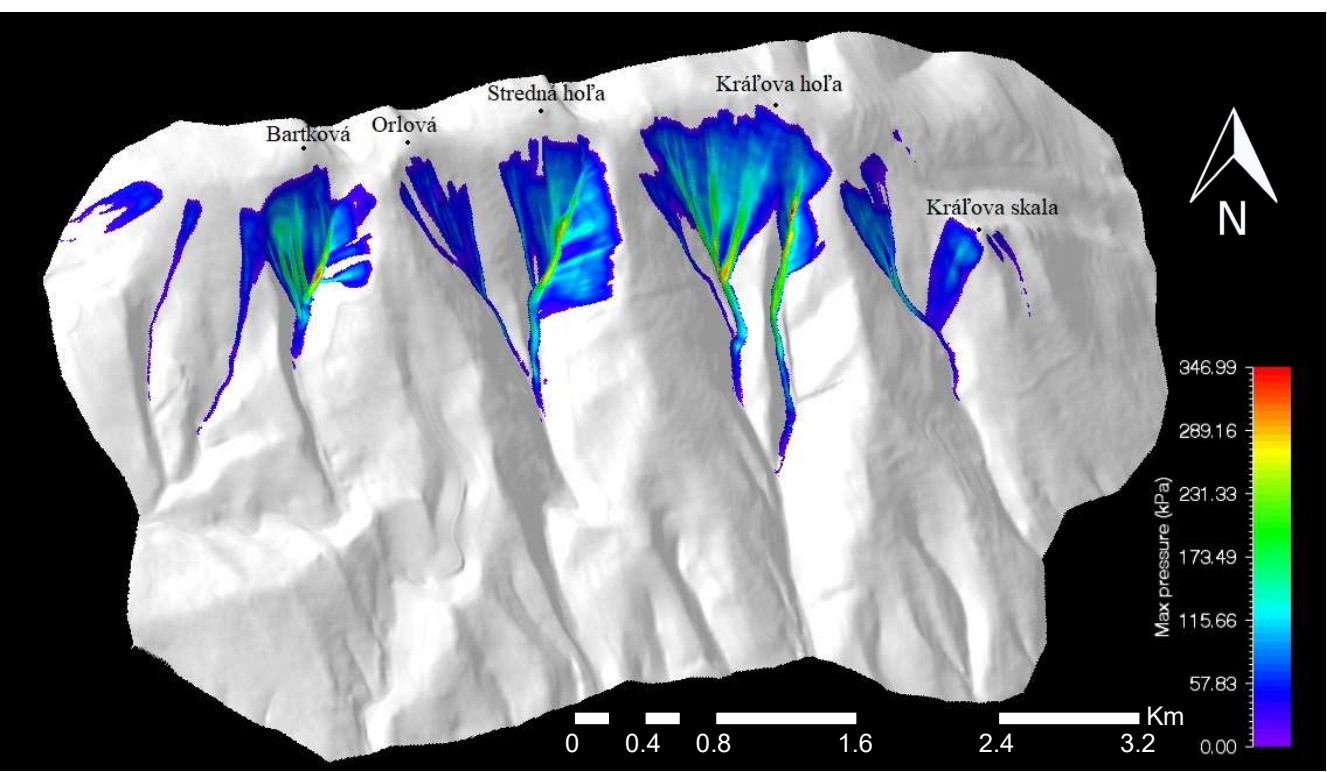

**Figure 6.** Maximum pressure exerted in avalanche fall (southern slope).

The maximum snow flow height (Figure 7) on the northern slope of the examined area was 13.54 m. This value was measured in the final part of the Holičná valley at an altitude of approximately 1500 m a.s.l. at the foot of the steep rock walls surrounding the glacial circuit Holičná. Relatively high values of this factor were also measured at the northeastern foot of the Stredná hoľa (1876 m a.s.l.), as well as in the area of steep gutter casts at two locations: (1) the northwest slope of Kráľova hoľa (1946 m a.s.l.) emerging into the valley Veľký Brunov and (2) the northeast slope of Bartková (1790 m a.s.l.) emerging into a nameless valley under the aforementioned elevation. The lowest values of this factor were recorded on steep grassy slopes, similarly to the southern side of the studied area, under the peak areas of the low Tatra ridge at altitudes from 1420 m a.s.l. to 1800 m a.s.l.

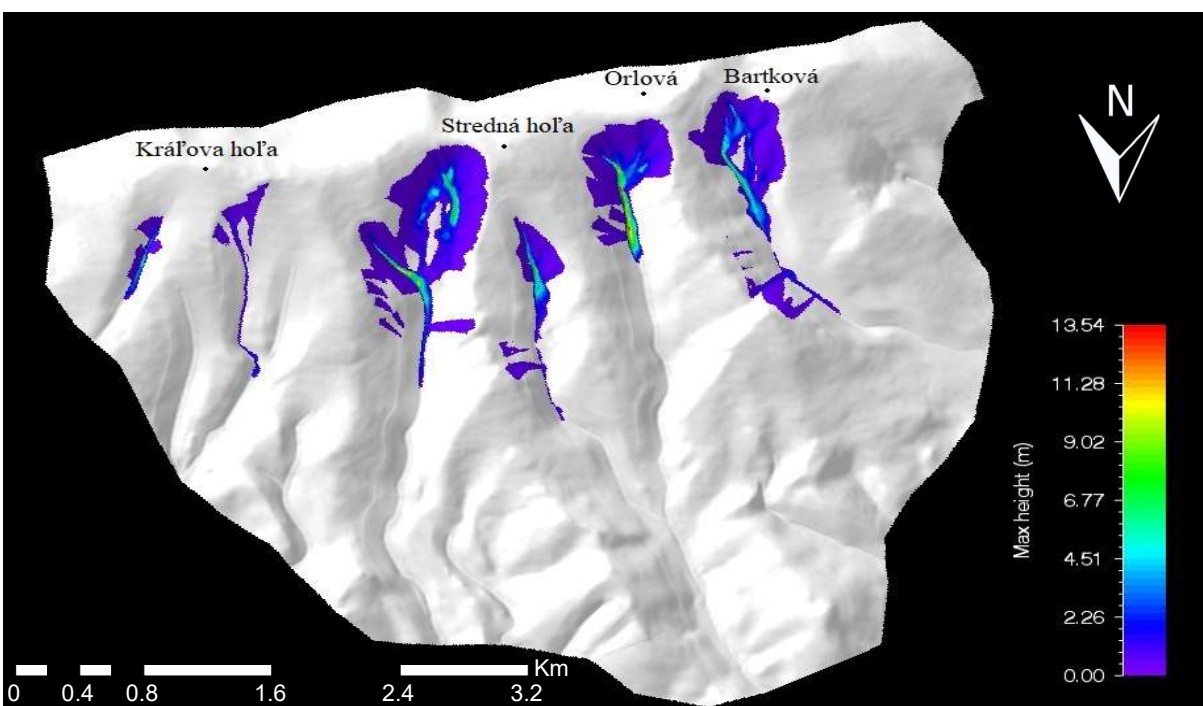

**Figure 7.** Maximum snow flow height (northern slope).

The average avalanche flow speed values (Figure 8) on the northern slope of the examined area range from 14 to 19 m/s. Maximum values were recorded in narrow steep gutter casts in the valleys of Veľký Brunov, Holičná, and a nameless valley under the northeastern slope of Bartková (1790 m a.s.l.) and on the edge parts of Brunov glacial cirques and a nameless cirque northeast of Bartková (1790 m a.s.l.) in the range from 19 to 28 m/s.

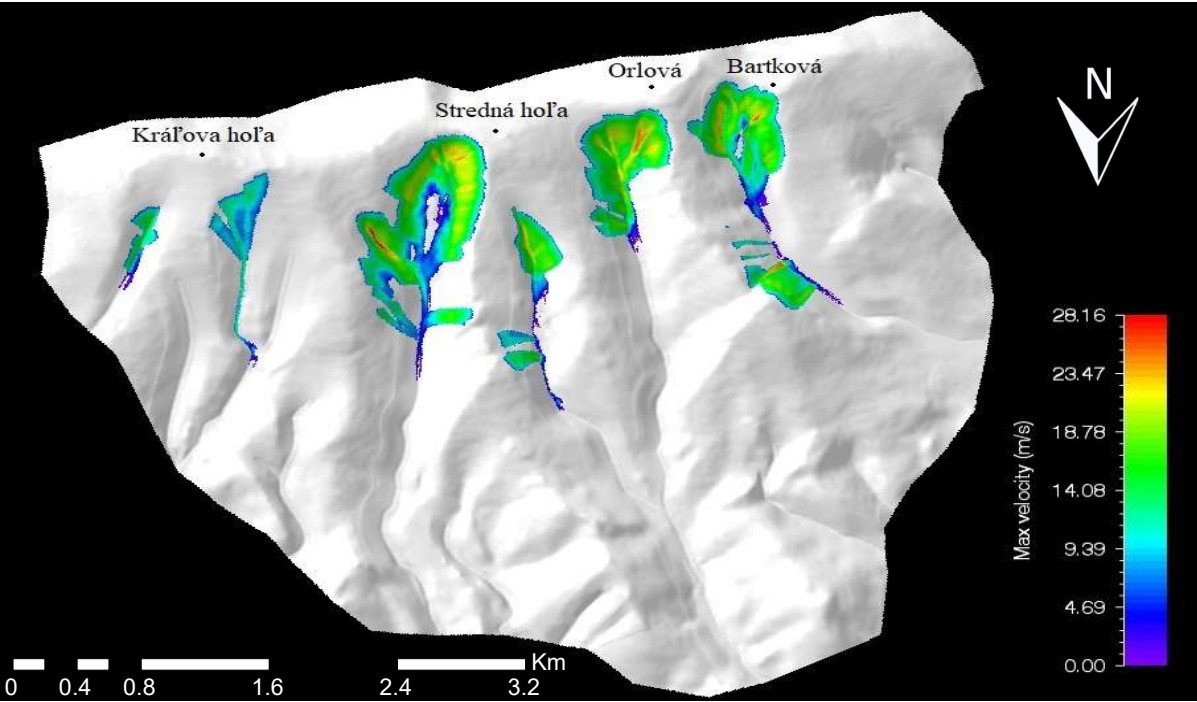

**Figure 8.** Maximum avalanche flow speed (northern slope).

The average value of the pressure exerted at avalanche fall (Figure 9) in the area of the valleys of Veľký Brunov, Holičná, and the nameless valley north of the Bartková (1790 m a.s.l.) is between 100 and 160 kPa. This indicator reaches its highest values at the mouth of the side gutter cast emerging into the Veľký Brunov valley (between 200 and 238 kPa).

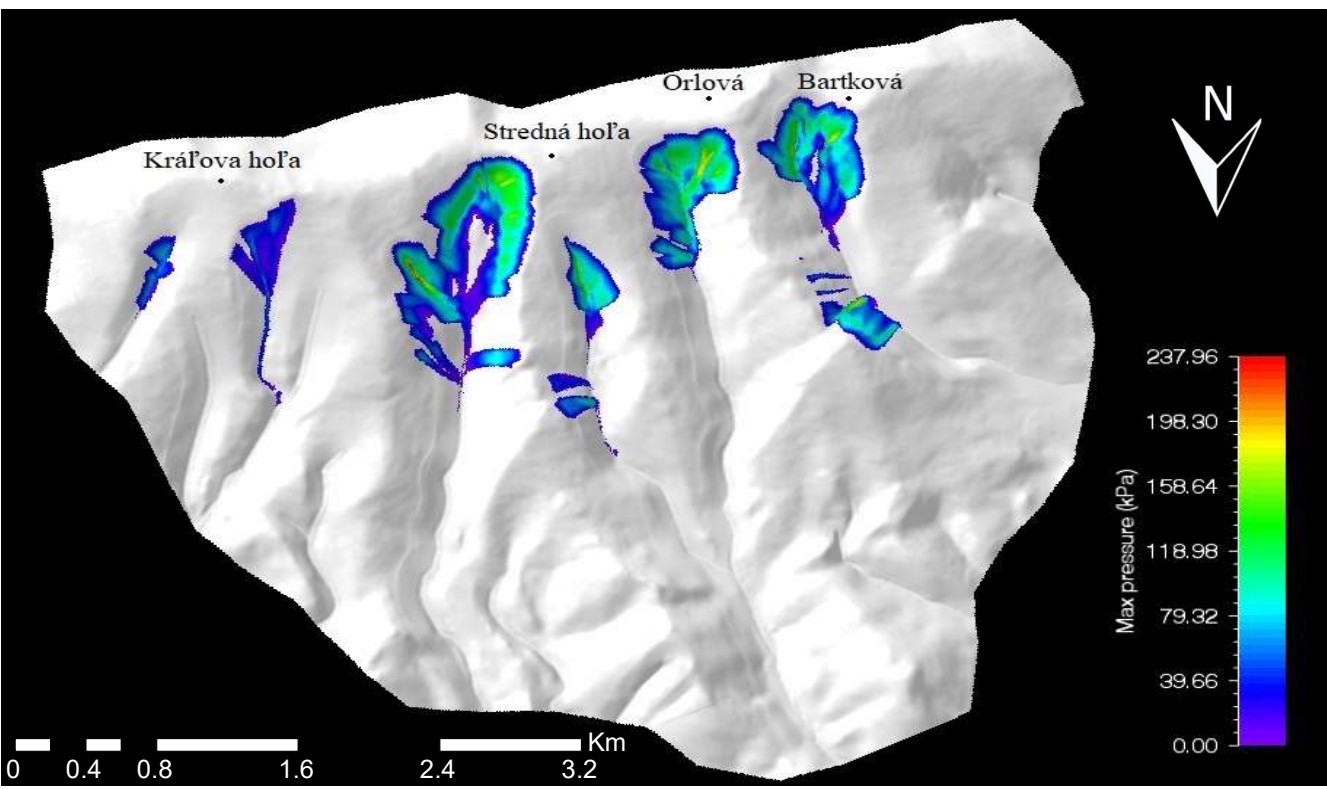

**Figure 9.** Maximum pressure exerted at avalanche fall (northern slope).

Figure 10 presents a layout of the avalanche cadastre, based on the results obtained from the RAMMS model. Most of the potential trigger areas occur in the peak parts of the low Tatra ridge, stretching from Ždiarske sedlo (1473 m a.s.l.) through Bartková (1790 m a.s.l.), Orlová (1840 m a.s.l.), Stredná hoľa (1876 m a.s.l.), and Kráľova hoľa (1946 m a.s.l.) to Kráľova skala (1690 m a.s.l.). The spatial expansion of trigger areas is directly determined by the slopes covered with grasslands, as well as by the existence of steep walls of glacial cirques located on the northeastern slopes of Bartková (1790 m a.s.l.), Orlová (1840 m a.s.l.) and Stredná hoľa (1876 m a.s.l.). In addition to the mentioned areas, we also recorded the possible occurrence of potential avalanches in steeper passages of the upper sections of the Čierny Váh, Hnilec, and Zubrovica watercourses. Most of the potential snow avalanches depicted display the features of gutter or surface-gutter avalanches, a typical sign of which is that the minimum width of the avalanche run-out measures only one third of the length of its trigger point. For this reason, the avalanche run-out path of this type of avalanche is relatively long and narrow, occurring especially in terrains with a significantly concave shape.

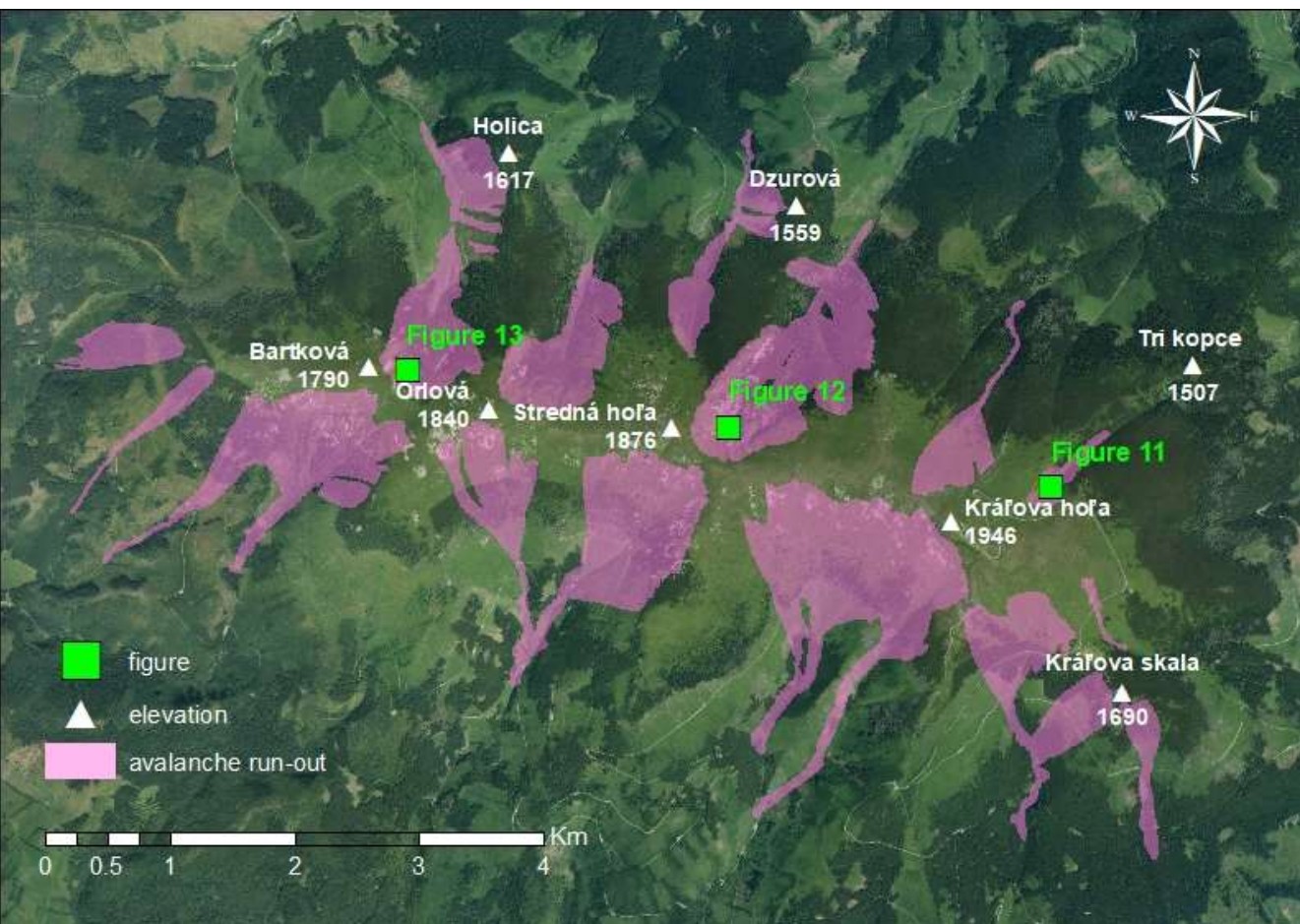

**Figure 10.** Layout of the avalanche cadastre based on results from the RAMMS model. Source orthophoto: The Geodesy, Cartography and Cadastre Authority of the Slovak Republic.

A total of fourteen interconnected avalanche run-out areas have been identified within the studied area. There are six of them at the northern foot of the ridge and at the southern foot there are eight. The largest complex of avalanche run-outs is located on the southern slope in Šumiacka dolina. Its area exceeds 2 km$^2$ and it has the highest width of the trigger zone reaching the value of 1793 m a.s.l. The second relatively large complex is located in Ždiarna dolina. Its area is 1.6 km$^2$. This fan-shaped complex is composed of two relatively separate parts (south of Orlová and Stredná hoľa), which merge into one unit in the lower accumulation part of Ždiarná dolina. The third larger complex on the south side of the ridge is the area at the top of the Lukačiková valley with an area of 1.2 km$^2$ and a trigger zone width of 1246 m. At the northern foothills of the studied area, there are some smaller avalanche run-outs—for instance on the NE slope of Kráľova hoľa hill (1946 m a.s.l.) (Figure 11). The largest complex of the avalanche run-outs at the northern foothills is located in the final section of the valley Veľký Brunov, with an area of 1.3 km$^2$ and the width of the trigger zone in the WE direction reaching 575 m (Figure 12). The second largest complex is in the nameless valley northeast of Bartková (1790 m a.s.l.) with an area of 0.87 km$^2$ located on the north side. It consists of two relatively isolated complexes connected by a thin line into one unit. The larger of these (directly below Bartková (1790 m a.s.l.) has a trigger zone in the WE direction with a width of 842 m (Figure 13). The second complex, located west of Holica (1617 m a.s.l.), has a trigger zone in the north-to-south direction with a width of 577 m.

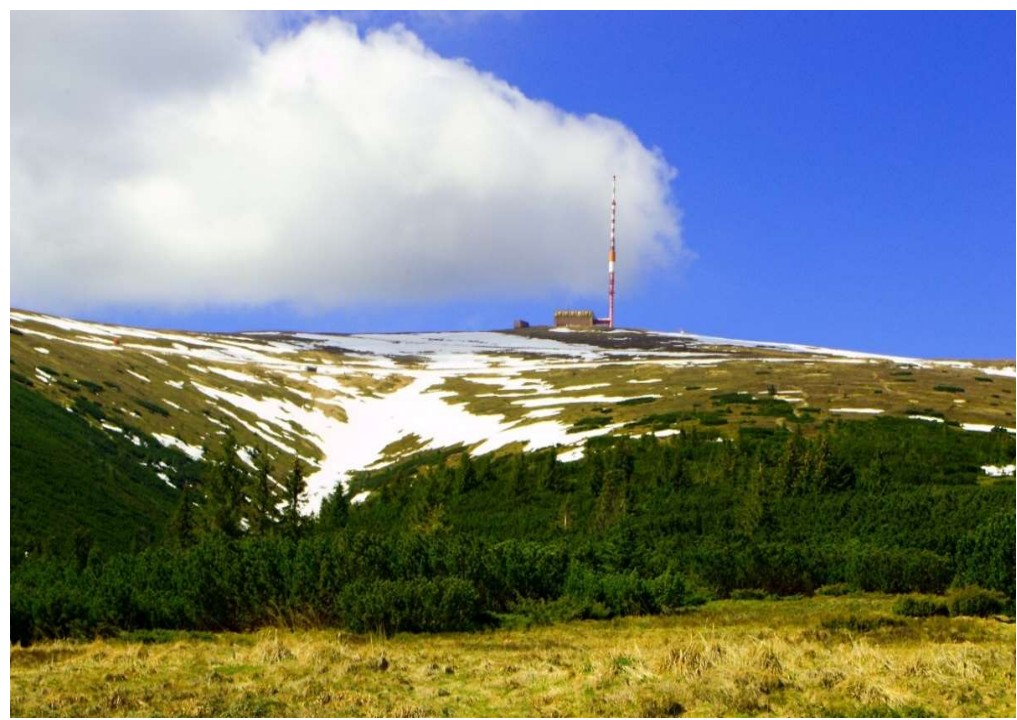

**Figure 11.** Identification of a trigger zone located in the Hnilec river spring area on the NE slope of Kráľova hoľa hill (1946 m a.s.l.).

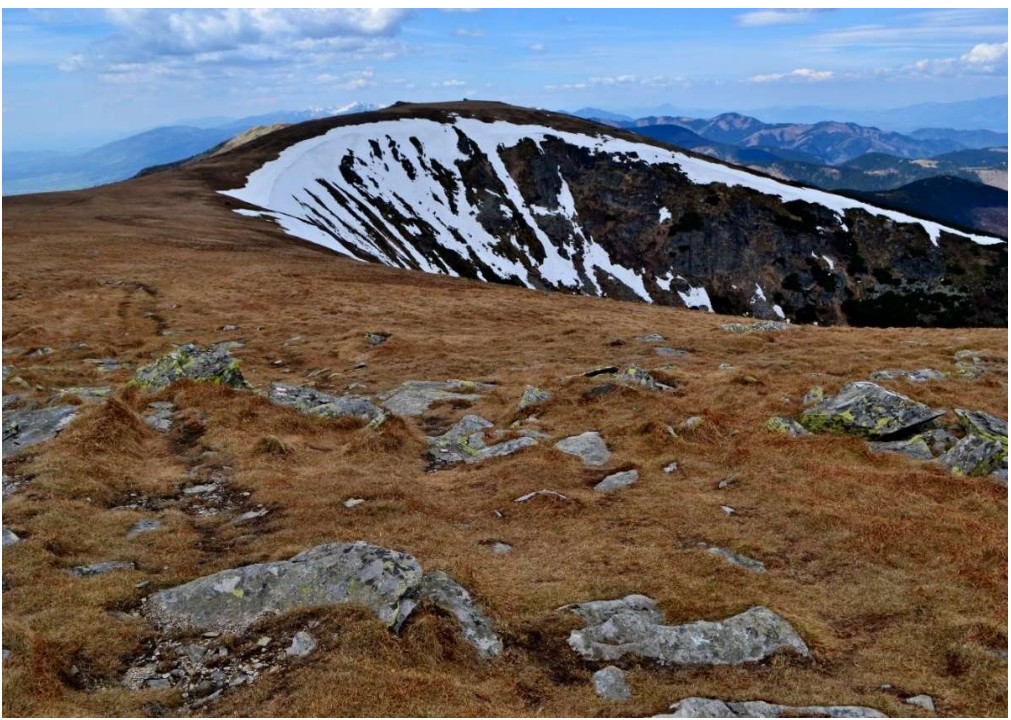

**Figure 12.** Identification of a trigger zone located along the edge of the glacial cirque Veľký Brunov.

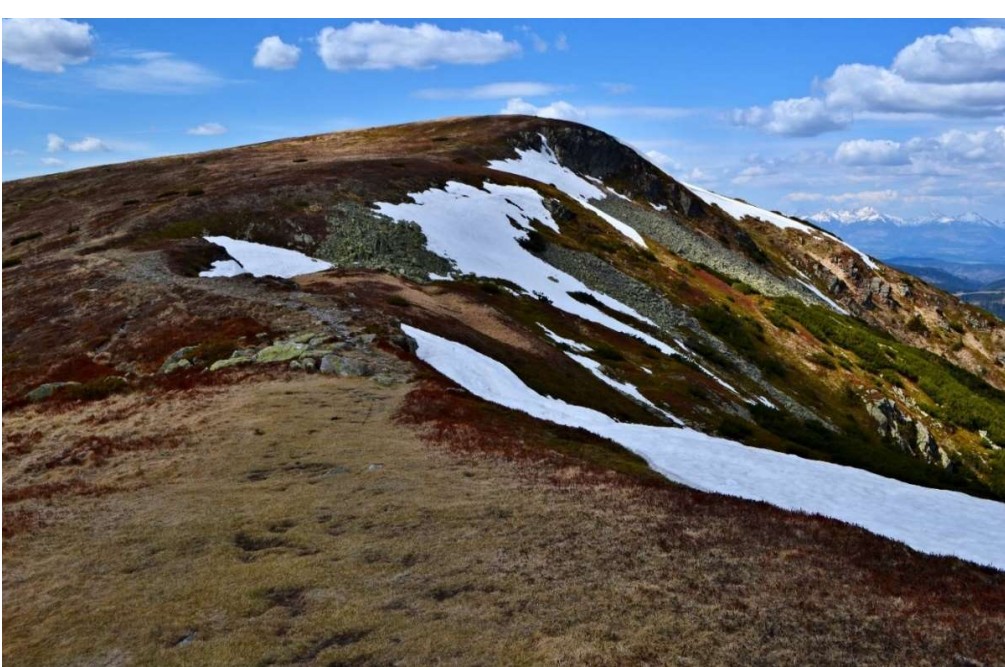

**Figure 13.** Identification of a trigger zone located along the edge of the glacial cirque below the top of Bartková.

## 4. Discussion

The eastern part of the Kráľova hoľa area of Low Tatras might not rank among the most avalanche-active areas of the Slovak Republic, but its typical clear cut grassy character in the peak ridge areas provides suitable conditions for the potential occurrence of snow avalanches. Based on the information provided by the Slovak Centre of Avalanche Prevention, there was only one recorded avalanche accident within this territory, in 1969, when a mass of snow triggered from the main ridge between Orlová (1840 m a.s.l.) and Stredná hoľa (1876 m a.s.l.) slid into the Holičná valley. The massive increase in the popularity of winter sports, combined with unsavory climatic and snow conditions, can increase the risk of potential of avalanches even on slopes that have not been affected by avalanche activity yet.

In the introductory part of this paper, we identified four research questions (hypotheses), which can now be reliably verified after conducting research on avalanche danger within the area of Kráľova hoľa.

The first hypothesis, concerning the more frequent occurrence of trigger zones and areas with a higher degree of avalanche risk on the northern slopes of the examined area, was partially refuted because, on the basis of the cartographic output (Figure 10), the total area of the trigger zones located on the north side of the examined territory was set at 1.2 km² and the area of trigger zones at the southern foot of the researched area was 1.4 km². A large extent of these zones on the southern slopes can also be attributed to the type of land cover, where, compared to the northern slopes, the range of grasslands (more prone to trigger zone occurrence with a higher degree of avalanche threat) is higher. It appears that the larger extent of glacial modelling of the northern slopes of the examined area does not have such a significant impact on the occurrence of trigger zones, which are instead conditioned by the cumulative impact of several factors. The intensity of glacial modelling and the resulting factors entering the assessment (slope, curvature, etc.) are more likely to be reflected in terms of the representation of territories with a high and very high degree of snow avalanche triggering risk. On the northern slope, the total area of these territories was 0.4 km² (29.71% of the total avalanche trigger zone extent on the northern slope) and on the southern slope it was 0.2 km² (15.73% of the total avalanche trigger zone extent on the southern slope).

The second hypothesis, dealing with the comparison of trigger zone extent on the examined territory compared to the multiple studies carried out within other Slovak mountains threatened by potential avalanche activity, has been confirmed. According to the elaborated cartographic output (Figure 3) the total area of trigger zones on the examined territory was calculated at the value of 2.6 km$^2$, in the whole region of the Kráľova hoľa area of Low Tatras it is 2.9 km$^2$. In the area of the eastern Tatras [75,76], the total area of trigger zones was set by Žiak [56] at 3 km$^2$. Trigger zones in the Malá Fatra mountains, which have become the subject of research for several avalanche experts [77–80] have a slightly higher area (5.1 km$^2$) according to Žiak [56]. The fourth mountain range with an area of trigger zones at 5.2 km$^2$ is, according to Žiak [56], Veľká Fatra, which was subject of avalanche hazards research by several authors [81–84]. The Ďumbier area of the Low Tatras belongs to the second position in the overall classification of areas threatened by avalanche activity in terms of the extent of trigger zones (11 km$^2$), according to Žiak [56]. The characteristics of the avalanche conditions of the studied area were the central theme of several scientific papers [85–88]. The territory of Western Tatras was clearly identified by Žiak [56] as the territory with the largest extent of trigger zones (33 km$^2$) within the territory of the Slovak Republic. The increased avalanche potential of this territory was also detected by the following author [89].

In the third hypothesis, it was necessary to verify the statement about the mutual combination of several factors entering into the avalanche risk assessment. This hypothesis was clearly confirmed through the research carried out, since in the complex modelling of the avalanche hazard of the Kráľova hoľa area, it was necessary to take into account the impacts of several factors: the altitude; the exposure; the horizontal and vertical curvature of the relief; the slope inclination; and the surface roughness. According to the degree of danger to the territory by triggering snow avalanches, the individual values of these factors were assigned to an appropriate value (Table 1). The mutual participation of the above-mentioned factors or other factors in mapping the potential avalanche hazards has been demonstrated in several leading publications [90–95].

The last research question (hypothesis) concerned the evaluation of the relevance of results obtained through the two-dimensional numerical simulation model RAMMS for the purpose of optimizing the potential use of land and preventing the risks posed by avalanche activity. This assumption has been confirmed, as pointed out by several authors in their multiple studies in different high mountain areas [96–103].

## 5. Conclusions

The examined area of the Kráľova hoľa area of Low Tatras represents a typical mountain range of the Carpathian arc stretching through the territory of eight states. In terms of the nature of the relief, there are traces of the quaternary glaciation with the occurrence of glacial cirques and moraines. Peaks with predominantly smoothly modeled ridges in the examined terrain represent suitable areas for potential avalanche formation. Endangerment of the area due to avalanche is influenced by several factors, while the area of trigger zones in the territory is larger on the southern slopes.

The submitted paper comprehensively evaluates the avalanche risk to the highest part of the Kráľova hoľa area of Low Tatras. It provides important data usable in several areas by means of the assessed potential for snow avalanche triggering and avalanche cadastre layout based on results from the RAMMS model.

This is the first work of its kind, which specifically deals with avalanche threat within this part of the Kráľova hoľa area of Low Tatras. Therefore, it represents a valuable contribution to the overall evaluation of avalanche risk present in the high mountains of Slovakia. It also provides a suitable basis for the subsequent assessment of the optimal land use and its proper management. The researched area is part of the Low Tatras National Park and represents an area intensively used by both the tourist and sports community. The impact of forestry on the overall nature of the area is also significant. This overview within the areas endangered by avalanche risk will enable the National Park Administration, in

cooperation with the Mountain Emergency Service, the State Forest, and other entities and organizations managing the examined area, to optimally manage the set of activities in this area with the aim of protecting the tourist and sports community in winter.

The influence of avalanches as a modeling factor on the overall geosystem of the studied area is also not negligible. As this area is part of a national park, all activities should be in accordance with the purpose and intention of the protected area status.

This paper can be extended to a larger area in the field of research and thus comprehensively evaluate the avalanche threat of geomorphological sub-region of the Kráľova hoľa area of Low Tatras. It is also desirable to focus on the impact of avalanches as a modelling factor affecting several components of the country's geosystem.

**Author Contributions:** Conceptualization, V.Č. and V.K.; methodology, V.K.; software, V.K.; validation, V.K., M.M. and M.J.; writing—original draft preparation, V.Č.; V.K., M.M. and M.J.; writing—review and editing, V.Č.; V.K., M.M. and M.J.; visualization, V.K. All authors have read and agreed to the published version of the manuscript.

**Funding:** This research was funded by KEGA agency, grant number 045PU-4/2022: Adrenalínový turizmus—dynamická oblasť v rozvoji cestovného ruchu.

**Institutional Review Board Statement:** Not applicable.

**Informed Consent Statement:** Not applicable.

**Data Availability Statement:** Not applicable.

**Acknowledgments:** The authors would like to thank Pavel Krajčí from the Mountain Rescue Service—Avalanche Prevention Center.

**Conflicts of Interest:** The authors declare no conflict of interest. The funders had no role in the design of the study; in the collection, analyses, or interpretation of data; in the writing of the manuscript, or in the decision to publish the results.

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
