# Peer review of "Avalanche Hazard Modelling within the Kráľova Hoľa Area in the Low Tatra Mountains in Slovakia"

_land, doi:10.3390/land11060766_

Round 1

Reviewer 1 Report

This work was the assessment of the avalanches in the Low Tatra Mountains in Slovakia. If the region is highly risky, it must be very important. However, the authors did not clearly noted that there were some accidents and it is risky. They said that "there was only one recorded avalanche accident within this territory". This really destroy the importance of this work. You should consider how to appeal your work.

The structure of the work is clear and understandable. The authors stated 4 hypothesis in introduction and answer these questions in discussion. On the contrary, some parts of the article are very ambiguous or not clear.

  1. What kind of avalanches do they have in the Tatra mountains? Which classification of avalanches do they use in their article?
  2. Not all mountain and region names are in the map. So it is very difficult to read results. Readers cannot imagine how the situation is. Additionally, the names in the maps are too small to recognize.
  3. The authors say that their aim is "comprehensive assessment of the avalanche". However, it is not clear if they would like to assess the origin/triggering of the flow or affected area of the flow (maybe both). In order to know the triggering, the authors should explain how they analyze the triggering area more clearly. Whole Table 1 is meaningless now. The authors should explain the concept and values in the table.
  4. The relationship between Figure 1and Figure 3 areas is unclear. What is a difference between "research area" and "study area"?

I would like to add some expressions problems.

  • The authors can think more on the digits. If you would like to compare two numbers, you should make the same digits.
  • Some expressions are weird in English. Also a space is necessary between number and units except “% (percent)” and “º (degree sign)”. You can ask English check for the next edition.
  • “Snow cover height” means the static condition of the snow. So it must be resplaced by “flow height”. The flow height is the result of the RAMMS simulation, and the snow cover height is the snow height before the snow starts to flow in my understanding.

Comments are also in a PDF file.

Author Response

Dear reviewer,

Thank you for your time and for any comments which help to improve our contribution. We hope that the adjustments we have made are in line with your suggestions and comments.

Reviewer 2 Report

Dear authors,

thank you for your study dealing with the avalanche danger modelling within the Low Tatra Mountains in Slovakia.  

The area of study is interesting and offer clues from the assessment of the avalanche hazard with the application of modelling techniques.

However, there are some aspects related to the aim and the layout of the manuscript not discussed and/or presented properly.

First, the study area is not clearly analyzed from the climatological and snow level point of view. The article could be improved to provide it scientific innovation and novelty since it follows the approach used in many other articles already published on all the main European mountain ranges.

Moreover, the layout of the manuscript is only to the modeling calculation; there is absolutely no presentation of a climatic framework at regional and local scale. The article lacks a detailed snow study in which are first highlighted the fundamental snow values (called decisive parameters) for the modeling calculation using the RAMMS software: the maximum historical height of fresh snow in 72 hours and the maximum height of the snow cover on the ground calculated in the respective areas of avalanche release, with a better reference to the Swiss method. According to this point, I suggest following the attached literature references: 

Oller, P.; Janeras, M.; de Buen, H.; Arnó, G.; Christen, M.; García, C.; Martínez, P. Using AVAL-1D to simulate avalanches in the eastern Pyrenees. Cold Reg. Sci. Technol. 2010, 64, 190–198.

Christen, M.; Bartelt, P.; Gruber, U. Numerical Calculation of Dense Flow and Powder Snow Avalanches; Swiss Federal Institute for Snow and Avalanche Research (SLF): Davos, Switzerland, 2010; p. 136

Salm, B.; Burkard, A.; Gubler, H.U. Berechnung von Fliesslawinen; eine Anleitung für Praktiker mit Beispielen; Mitteilungen des Eidgenössischen Institutes für Schnee und Lawinenforschung: Davos, Switzerland, 1990; p. 37.

I suggest revise the manuscript to better describe the climatic features and the provided assessment of avalanche hazard. In detail, please introduce a new chapter named “Climatic setting of the area” concerning the climatic framework and the nivologic and nivometric. Please add a better description, in the Discussion Section, about the results of your analysis.

Please follow as examples and add some recent literature reference:

Fazzini, M.; Cordeschi, M.; Carabella, C.; Paglia, G.; Esposito, G.; Miccadei, E. Snow Avalanche Assessment in Mass Movement-Prone Areas: Results from Climate Extremization in Relationship with Environmental Risk Reduction in the Prati di Tivo Area (Gran Sasso Massif, Central Italy). Land 2021, 10, 1176. https://doi.org/10.3390/land10111176

Statham, G., Haegeli, P., Greene, E. et al. A conceptual model of avalanche hazard. Nat Hazards 90, 663–691 (2018). https://doi.org/10.1007/s11069-017-3070-5

Fischer, J., Kofler, A., Fellin, W., Granig, M., & Kleemayr, K. (2015). Multivariate parameter optimization for computational snow avalanche simulation. Journal of Glaciology, 61(229), 875-888. doi:10.3189/2015JoG14J168

Then, I would suggest you revise the Figures, because, according to this layout, it is very difficult to read and discriminate all the reported elements.

Finally, please note my comments and suggestions in the attached PDF document.

Author Response

(The authors gave the same response as above.)

Round 2

Reviewer 1 Report

Dear Authors,

Please refer my word file.

Author Response

Dear reviewer,

Thank you for your time and for any comments whitch help improve our contribution. We hope that the adjustments we have made are in line with your suggestions and comments.

Reviewer 2 Report

Dear authors,

thank you for the new version of the manuscript. It is widely updated in comparison to the previous one.

However, there are some aspects related to the aim and the layout still not discussed and/or presented properly.

Finally, please note my minor comments and suggestions in the attached PDF document.

Author Response

(The authors gave the same response as above.)
